# Effect of Seedling Provenance and Site Heterogeneity on *Abies cephalonica* Performance in a Post-Fire Environment

Kostas Ioannidis [1],*, Marianthi Tsakaldimi [2], Katerina Koutsovoulou [3], Evangelia N. Daskalakou [4] and Petros Ganatsas [2],*

[1] Laboratory of Silviculture, Forest Genetics and Biotechnology, Institute of Mediterranean & Forest Ecosystems, Hellenic Agricultural Organization "Demeter", 11528 Athens, Greece

[2] Laboratory of Silviculture, Department of Forestry and Natural Environment, Aristotle University of Thessaloniki, 54124 Thessaloniki, Greece; marian@for.auth.gr

[3] Green Fund, Ministry of Environment and Energy, 14561 Kifisia, Greece; kkoutsovoulou@gmail.com

[4] Laboratory Forest Management & Forest Economics, Institute of Mediterranean & Forest Ecosystems, Hellenic Agricultural Organization "Demeter", 11528 Athens, Greece; edaskalakou@fria.gr

\* Correspondence: ioko@fria.gr (K.I.); pgana@for.auth.gr (P.G.)

**Abstract:** Reforestation constitutes a challenge in post-fire ecosystem restoration, although there are limitations such as species and genotype selection, planting and management design, and environmental conditions. In the present study, the basic issue is the longevity of *Abies cephalonica* Loudon—the Greek fir seedlings planted extensively in Parnitha National Park (Central Greece), located near the metropolitan city of Athens, following the large-scale wildfire of 2007. Seedling performance was assessed for a 3-year monitoring period (2013–2015) through the establishment of 8 permanent transects, including 400 seedlings at the burned, reforested sites. According to the long-term reforestation project, two seedling provenances were used: (a) from Mt. Mainalon (South Greece, Vytina provenance) and (b) the local one from Mt. Parnitha. Both provenances showed a relatively successful survival rate reaching, in average, 73.8%, with the first summer after planting being crucial for seedling survival. The overall mean seedling height was 39.2 ± 1.1 cm, with a mean crown diameter of 47.3 ± 1.4 cm in the last monitoring survey. Although Parnitha seedlings seem to perform better in terms of growth, seedling performance in both provenances was affected by reforestation site characteristics, mainly altitude and aspect. Approximately one third of seedlings exhibited damage in their crown architecture (29.8%), while apical bud damage was less extensive (12.2%) in the final field measurement. Data indicate that seedling performance has proved to be quite promising for post-fire restoration, although long-term monitoring data should be considered.

**Keywords:** Greek fir; reforestation; species conservation; provenance; seedling survival and growth; ecological restoration

## 1. Introduction

Among the extensive types of forest ecosystems worldwide, there are some that appear locally, with a restricted distribution in small areas mainly due to evolutionary processes. These ecosystems are considered to be inherently vulnerable because of their usually narrow environmental envelopes, their geographically restricted distribution, and the fact that many of them appear to be near climatic thresholds [1,2]. As regards global warming, the potential reduction in available moisture and extreme drought conditions could be the biggest future threat, either alone or combined with wildfires. Thus, special attention should be given to these issues by the scientific community in order to secure the existence and sustainability of these important ecosystems [3].

Tree natural regeneration is an essential process in forest ecosystems to ensure the persistence and resilience of forest stands when subjected to various disturbances, especially fires, and should contribute to a gradual process of recovery of the structure, function,

and composition of the pre-disturbance ecosystem [4,5]. Successful seed germination and seedling survival and growth are the major factors for in situ species conservation [6]. However, when naturally regenerating forests are unlikely to recover or face difficulties in their self-renewing process, an active restoration approach with enrichment plantings may be required to increase the abundance of poorly dispersed species or to protect endangered species and secure their long-term conservation [4,7–9].

*Abies cephalonica* Loudon (commonly known as Greek fir) is an endemic, important fir species with extremely restricted distribution found only in the southern part of the Balkan peninsula, and specifically in Southern and Central Greece [10–13]. It is a Mediterranean mountainous tree species that grows in relatively high altitudes, usually between 800 and 2000 m a.s.l. [11]. Greek fir is shade-tolerant, with relatively high demand for soil moisture, an environmental variable considered as the crucial limiting factor for the species' growth and survival [14]. Due to its abovementioned ecophysiological attitudes, its regeneration, as is the case with many other fir species, in forest management systems is obtained inside an interior forest environment, under the protection of canopy cover of mother trees, applying shelterwood cuttings [15–17]. This partially shaded environment protects the sensitive young fir seedlings from intensive sunlight and additionally creates favorable microsoil conditions, such as soil moisture, for the recruitment and growth of young seedlings [18].

Due to climate change and summer drought episodes that have increased in the relatively more humid and colder regions of Greece [19], a shift of wildfires toward higher altitudinal ecosystems, including fir ecosystems, has been observed [20–22]. This phenomenon increases fire danger and, hence, the displacement risk of these mountainous fir ecosystems since they have not developed fire-adapted mechanisms. Greek fir does not produce serotinous cones and does not maintain a seed bank to secure its conservation after a wildfire [21]. Findings from previous studies have concluded that the post-fire regeneration of *A. cephalonica* in Mt. Parnitha, three years after the wildfire, tends to be zero despite the existence of nearby unburned fir stands [6]. However, Raftoyannis and Spanos [23], who studied the post-fire regeneration of the Greek fir in a much more humid environment, found that it depended mainly on the distance from the seed source, i.e., from the remnant stands or the border of the unburned zone. Thus, in order to ensure the conservation and endurance of Greek fir forests following various disturbances, effective restoration techniques should be developed, following the general rules of reforestation and aiming at the optimization of carbon sequestration, biodiversity conservation, and livelihood benefits [24]. These techniques concern the selection of suitable site-specific provenances, setting specific standards for seedling production, and appropriate site-preparation techniques, especially those addressing post-fire environmental conditions. Special care should also be given to the fact that Greek fir ecosystems are mostly distributed within protected areas (e.g., European ecological network Natura 2000). Thus, the provenance (origin) of the plant reproductive material intended for reforestation projects in protected areas is of high importance, in order to avoid any genetic degradation of the local fir populations. Taking into consideration the fact that wildfires were not a common phenomenon in fir ecosystems in the past, many of the above issues concerning post-fire restoration have not been studied at all. Scarce studies have examined the issues pertaining to Greek fir, such as the influence of seedling height and site parameters (shade, competitive vegetation, bedrock, weather conditions) on survival and growth of outplanted *A. cephalonica* seedlings [25,26]. However, most of the factors affecting reforestation success in a post-fire environment have not been thoroughly studied. Although the diversification of genetic plant material in reforestations is considered a promising strategy to promote forest adaptation to climate change [27], no research so far has studied the survival and growth of different provenances of *A. cephalonica* in a Mediterranean post-fire environment.

In the present study, we document, for the first time, the effect of seedling provenance as well as site heterogeneity (altitude, aspect, slope, bedrock–soil parent material) on a 3-year monitoring reforestation success (survival and growth) of Greek fir seedlings in the post-fire environment of Parnitha National Park (Attica, central Greece). Furthermore,

seedling damage on crown architecture and apical bud, probably attributed to wildlife activity (local population of red deer, *Cervus elaphus*-Cervidae) in the post-fire environment was assessed.

## 2. Materials and Methods

### 2.1. Study Area

Parnitha National Park is located in Mt. Parnitha, the highest (1413 m a.s.l.) mountain near the Athens metropolitan area (Attica region, Greece, Figure 1). The core zone of the Park comprises the high peaks of Parnitha (circa 3800 ha), and most of that area (90%) is covered with pure, mature *A. cephalonica* stands which extend from 800 m a.s.l. up to the summit of the mountain. There are a few populations of the species worldwide, and thus, the forest of Parnitha comprises an important biodiversity area; for that reason, the site has been characterized as a National Park and has been included in the European Natura 2000 network [6,13,28]. This forest could be considered an ecotype, genetically differentiated and adapted to adverse and unfavorable conditions [12]. Geologically, the area bedrock consists of sedimentary rocks, mainly flysch and limestone as parental material [26]. Soil is generally shallow with low fertility [29]. The climate of the area differs considerably from that of the rest of Attica; at 1000 m elevation, the average annual rainfall is 822 mm, the annual snow level is 120 cm, the average annual relative humidity is 77%, the annual rainfall days are 70, snowy days are approximately 33, and the average annual temperature is 11 °C [6].

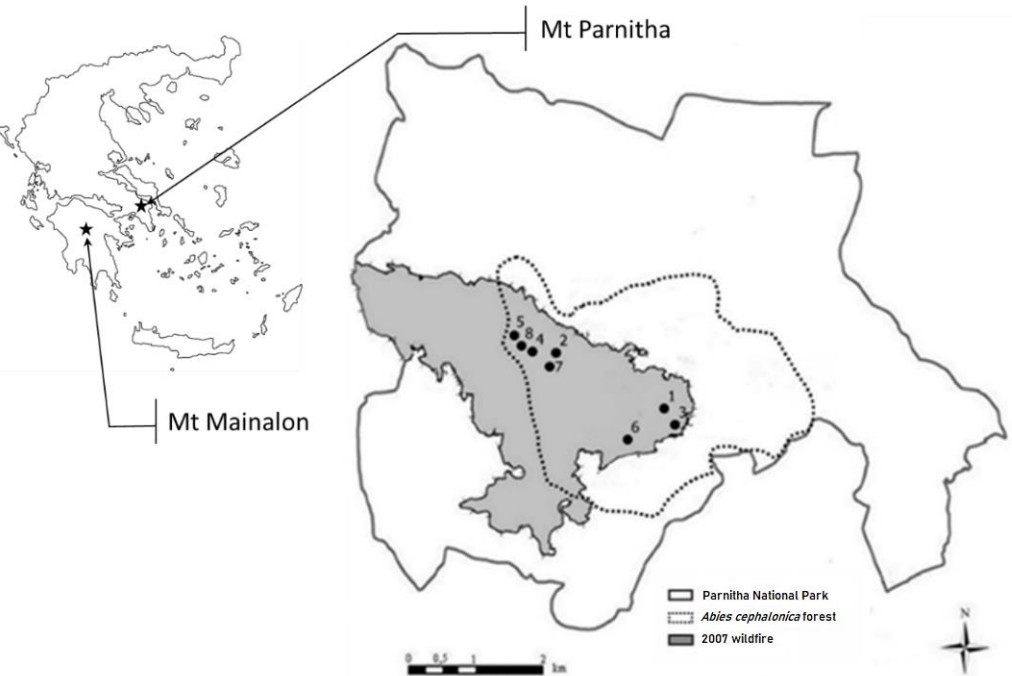

**Figure 1.** Monitoring transects established (autumn 2012) on the burned and reforested sites of *Abies cephalonica* forest (adapted from Daskalakou et al., 2019).

During the summer of 2007, an unpredictable crown wildfire [30] of high-severity [28] completely burned and destroyed a great part (approx. 60%) of the fir forest, resulting in a considerable degradation of the *A. cephalonica* habitat [6,13,25,26,31]. Due to the species' extremely limited regeneration potential after fire [6], a long-term reforestation project was carried out the next year by the Forest Service of Parnitha in order to restore the burned fir ecosystem. In the present study, "reforestation" concerns the establishment of forest plantations on temporarily unstocked lands that are considered to be "forest" [32]. Plantings with three-year-old, containerized *A. cephalonica* seedlings were conducted. Two provenances were used: (a) seedlings from Mt. Mainalon (South Greece, Vytina provenance,

Figure 1) and (b) seedlings from Mt. Parnitha (Parnitha provenance). Vytina seedlings were planted in 2008 [6,25,26], and Parnitha seedlings, the local provenance, in 2011 and 2012. In order to mitigate the differences over the planting years on the reforestation success follow-up, we have chosen the approach of calculating height and crown diameter change (growth), expressed on a percentage basis among monitoring periods [33]. Moreover, we used appropriate methodologies in order to test the hypothesis that there are differences in survival and growth between provenances [27,34–36].

### 2.2. Field Data Collection

The reforestation data on *A. cephalonica* seedling performance in the field were gathered during a 3-year monitoring period, 2013–2015 [37]. Seedling performance and survival were assessed through the establishment of eight (8) permanent transects at the burned and reforested sites of the National Park in autumn 2012 (Figure 1). Each transect circa 100 m × 2 m included fifty (50) seedlings. In total, 400 seedlings were labeled with a unique identity number. Seedling survival (%), height (cm), and crown diameter (cm) were recorded twice per year, at the beginning of the growing season and after the summer dry period, in late spring (May) and in fall (October), respectively. In each transect, geographical coordinates, altitude, slope, aspect, and type of bedrock were recorded (Table 1). We classified the transect elevation range into 4 altitudinal zones, i.e., 926–958 m, 959–991 m, 992–1024 m, and 1025–1057 m, in accordance with Sturges' class interval formula [38]. The recorded aspects in all sampled transects were North (N), South (S), and East (E); in 2012, there were no *A. cephalonica* reforested sites in West (W) aspects–transects. The recorded slopes were categorized into three classes, <10%, 10–30%, and >30%.

**Table 1.** Characteristics of the reforested sites with *A. cephalonica* seedlings in Parnitha National Park.

| | Transect Local Name | Seedling Provenance | Aspect | Altitude (m) | Slope Class (%) | Bedrock Type * | Coordinates | |
|---|---|---|---|---|---|---|---|---|
| | | | | | | | Latitude | Longitude |
| 1 | Agia Triada | Vytina | South | 1041 | 10–30 | F | N38°09′17.1″ | E23°43′39.0″ |
| 2 | Perdikovrahos | Vytina | North | 1002 | 10–30 | F | N38°10′05.2″ | E23°41′40.4″ |
| 3 | Casino | Vytina | South | 1056 | 0–10 | L | N38°09′03.0″ | E23°43′51.4″ |
| 4 | Panos | Parnitha | North | 964 | 10–30 | F | N38°10′06.4″ | E23°41′14.5″ |
| 5 | Lakka Zareli | Parnitha | North | 926 | >30 | F | N38°10′20.0″ | E23°40′54.1″ |
| 6 | Sanatorio | Parnitha | South | 1035 | 0–10 | L | N38°08′50.4″ | E23°42′59.6″ |
| 7 | Gaidourovrysi | Parnitha | East | 1024 | 10–30 | F | N38°09′53.3″ | E23°41′33.7″ |
| 8 | Korakovrahos | Parnitha | East | 959 | 0–10 | L | N38°10′11.6″ | E23°41′02.3″ |

\* L, limestone; F, flysch.

The presence of the apical bud and the crown architecture conditions i.e., damaged or intact crown, was also monitored in order to determine the crown development in relation to environmental factors (transect characteristics). Seedling crown diameter was calculated by the average of the individual maximum and minimum seedling crown diameter, which was measured in the field during the study period.

### 2.3. Statistical Analysis

A binomial logistic regression (forward, backward, and hierarchical) was performed to predict the probability of seedling survival, which was considered as the dichotomous dependent variable, using as predictor variables the origin of the seedlings (provenance), aspect, inclination, and elevation. Percentage change (*p*) in height and crown diameter of the planted seedlings was calculated by the equation: $p = (M_2 - M_1) * 100 / M_1$, where $M_1$ and $M_2$ are height or crown diameter measurements in the autumn of 2012 and 2015, respectively. Data in percentages were subjected to appropriate log or square root transformation before statistical analysis and were transformed back to percentages.

Several models were tested in order to select the appropriate one that has a good fit to the data. We also applied several general mixed models testing the characteristics of the

monitoring transects as well as their interactions' impact on the percentage change in the height and crown diameter of the planted seedlings. The independent variables were the provenance (nominal variable: Vytina/Parnitha), the aspect (N, S, E), the altitudinal zone (<926–958 m, 959–991 m, 992–1024 m and 1025–1057 m), the slope class (0–10%, 10–30% and >30%), and the bedrock type (flysch and limestone).

Analysis of seedling growth was performed applying several general mixed models in order to test the significance of the measured independent variables, and we isolated the significant variables. All effects were considered random. The following mixed model was used in the analysis:

$$Gp_{ijmk} = \mu + o_i + a_j + h_m + e_{ijmk} \tag{1}$$

where $Gp_{ijmk}$ is the percentage change in height or crown diameter (growth) of the $k^{th}$ seedling, $i^{th}$ seedling provenance, $j^{th}$ transect aspect, and $m^{th}$ altitudinal zone as a dependent variable, $\mu$ is the fixed population mean percentage change in seedling height or crown diameter of all individuals, $o_i$ is the random effect of the $i^{th}$ origin, $a_j$ is the random effect of $j^{th}$ aspect, $h_m$ is the random effect of $m^{th}$ altitude, and $e_{ijmk}$ is the random residual error of $k^{th}$ seedling, $i^{th}$ seedling provenance, $j^{th}$ transect aspect, and $m^{th}$ altitudinal zone.

Analysis of variance (ANOVA) was used to assess the difference between the measured variables. When ANOVA indicated a significant F-value, Duncan's test at $p < 0.05$ was performed to compare the means. Moreover, *t*-test was used to determine if the means of two sets of data were significantly different from each other. All statistics were performed using SPSS v.20 software for Windows (IBM SPSS Statistics 2011, IBM Corp. New York, NY, USA).

## 3. Results

### 3.1. Overall Seedling Performance

At the end of the monitoring period, i.e., 2015, 73.8% of the total seedlings had survived. The overall mean seedling height and crown diameter were 39.2 ± 1.1 and 47.3 ± 1.4 cm, respectively. At the preliminary monitoring survey (autumn 2012), the relevant values were 20.7 ± 0.6 and 20.3 ± 0.8 cm, respectively. The 3-year height and canopy diameter growth, expressed as a percentage of the initial values, was 85.3% and 158.6%, respectively (Table 2). In autumn 2015 (final monitoring period), the percentage of seedlings with intact crowns accounted for 70.2%, and those with apical bud represented 87.8%.

**Table 2.** The *Abies cephalonica* seedling average performance for the variables studied in the last monitoring period (autumn of 2015) in reforestations of Parnitha National Park.

| Dependent Variable | Number of Seedlings | Mean (%) | * S.E. (%) |
|---|---|---|---|
| Survival | 400 | 73.8 | 2.20 |
| Overall height growth | 295 | 85.3 | 3.99 |
| Overall crown growth | 295 | 158.6 | 7.77 |
| Crown condition intact | 295 | 70.2 | 2.67 |
| Apical bud presence | 295 | 87.8 | 1.90 |

* S.E: standard error.

### 3.2. Analysis of Seedling Survival

The provenance significantly affected the survival rate of the planted *A. cephalonica* seedlings (Figure 2a); the majority of the nonsurviving seedlings were recorded after the summer (drought) period (Figure 2b). During the 3-year monitoring period, Vytina seedlings exhibited a continuously significantly greater survival rate than those of Parnitha. In autumn 2015 (last survey), the survival rate of Vytina seedlings was 87.3%, while that of Parnitha was 65.6%.

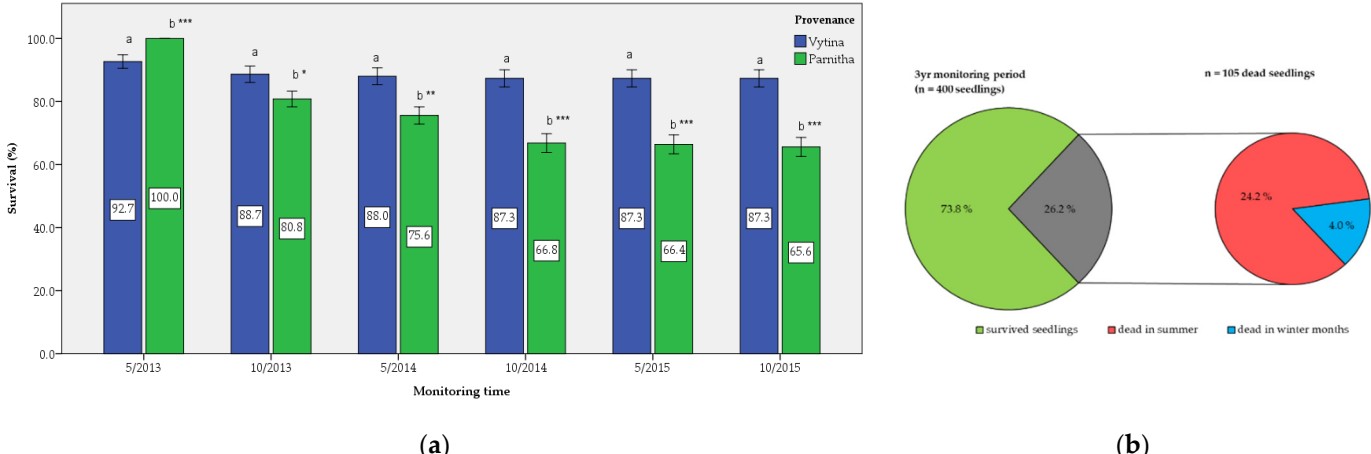

(**a**)                                                                                                              (**b**)

**Figure 2.** *Abies cephalonica* seedling survival (%) during the 3-year monitoring period 2013–2015 (**a**); overall seedling mortality (%) recorded in the summer and winter periods; (**b**). Time zero, i.e., survival 100%, is autumn 2012; monitoring took place twice annually, in spring (May) and in autumn (October). Survival means in the same monitoring time followed by different letters are statistically different at $p \leq 0.05$ *, $p \leq 0.01$ **, and $p \leq 0.001$ *** according to Duncan's test.

In addition to the origin and using the aspect, inclination, and elevation exposure as additional predictor variables, the selected binomial logistic regression model adequately fitted the data as the logistic regression model was statistically significant ($\chi^2(3) = 48.196$, $p \leq 0.000$), and the Hosmer and Lemeshow test resulted in $\chi^2 = 2.436$ ($p \leq 0.656$). The model correctly classified 77.3% of cases. The explained variation in the dependent variable based on our model ranges from 11.4% to 16.6%, depending on whether we reference the Cox and Snell $R^2$ or Nagelkerke $R^2$ methods, respectively. From the analysis, we can see that provenance ($p \leq 0.026$) and aspect ($p \leq 0.000$) significantly enhanced the model/prediction (Figure 3).

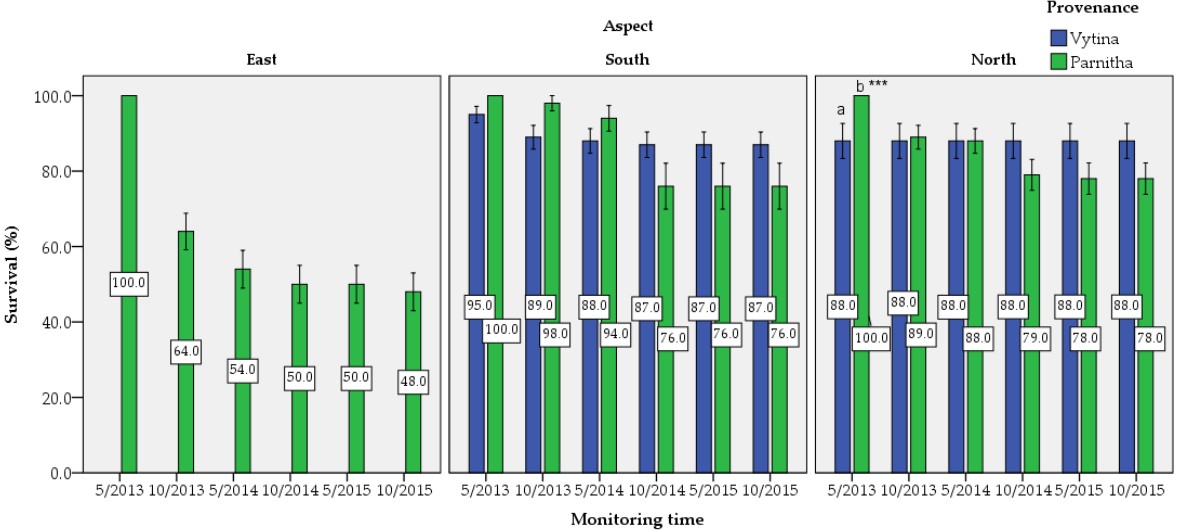

**Figure 3.** *Abies cephalonica* seedling survival (%) during the 3-year monitoring period (2013–2015) in relation to their provenance and transect aspect. Survival means in the same monitoring time followed by different letter are statistically different at $p \leq 0.001$ *** according to Duncan's test.

Concerning the effect of aspect, seedling survival in east-facing transects was found to be relatively low, less than 50% (i.e., 48%) at the end of the monitoring period (Figure 3). The first year after planting, the mortality rate had reached 36%, and after two years, half of the seedlings had not survived. By contrast, seedling survival was high in north- and south-

facing transects, presenting average final survival rates of 83% and 81.5%, respectively, without significant differences between the two aspects.

### 3.3. Analysis of Seedling Growth

#### 3.3.1. Seedling Height

The analysis of variance revealed that seedling provenance ($p \leq 0.01$), aspect ($p \leq 0.05$), and altitude ($p \leq 0.01$) significantly affected height growth during the 3-year monitoring period (Table 3), although not all origins are represented in all experiments.

**Table 3.** ANOVA table for planted *Abies cephalonica* seedling height growth, according to GLM: $Gp_{ijmk} = \mu + o_i + a_j + h_m + e_{ijmk}$.

| Source of Variation | df | Mean Square | F | Sig. |
|---|---|---|---|---|
| Provenance | 1 | 35,633.20 | 7.953 | ** |
| Aspect | 1 | 22,592.14 | 5.042 | * |
| Altitude | 2 | 24,441.50 | 5.455 | ** |

** $p \leq 0.01$, * $p \leq 0.05$.

The seedling height (cm) and the mean percentage change in height (HGP) in relation to seedling provenance during the 3-year monitoring period are presented in Figure 4. During this period, the height percentage change of Parnitha seedlings in most cases was greater than that of Vytina ones. It is also observed that even though seedling mortality occurred mostly during the summer period, seedling growth continues not only during summer, but almost during the whole year. At the end of the monitoring period (autumn 2015), the overall height growth percentage was significantly higher in Parnitha seedlings (92.7%) compared to those originating from Vytina (75.4%), (Table 4).

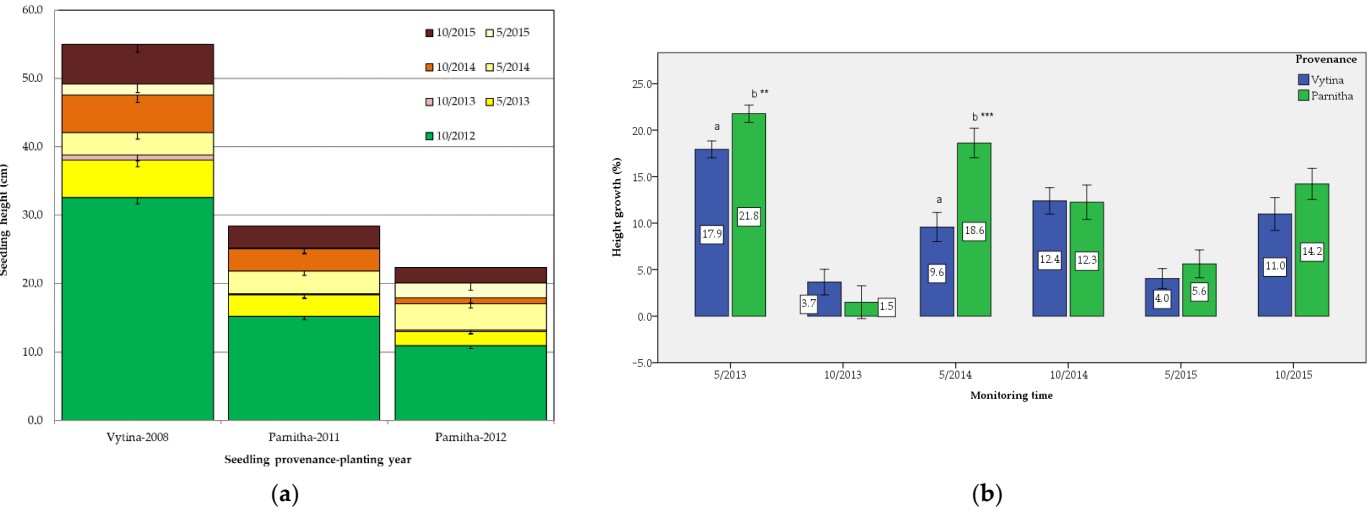

(**a**)          (**b**)

**Figure 4.** Height (cm) of planted *Abies cephalonica* seedlings (**a**) and the overall mean percentage change in height (%) (**b**) during the 3-year monitoring period in relation to seedling provenance. Height growth means in the same monitoring time followed by different letters are statistically different at $p \leq 0.01$ ** and $p \leq 0.001$ *** according to Duncan's test.

**Table 4.** The overall mean percentage change in height (HGP) of planted *Abies cephalonica* seedlings at the end of the monitoring period (autumn 2015), in relation to seedling provenance, transect aspect, and altitude. Different letters represent statistically significant differences within the same column ($p \leq 0.05$, Duncan's test).

| Origin | N | | HGP |
|---|---|---|---|
| Parnitha | 164 | | 92.7 [a] |
| Vytina | 131 | | 75.4 [b] |
| | **Parnitha** | **Vytina** | |
| | **Aspect** | | |
| | N | N | |
| North | 78 | 44 | 94 [a] |
| East | 48 | - | 91.1 [ab] |
| South | 38 | 87 | 74.5 [b] |
| | **Altitude** | | |
| | N | N | |
| 926–958 | 39 | - | 74.7 [b] |
| 959–991 | 57 | - | 98.5 [a] |
| 992–1024 | 30 | 44 | 98.8 [a] |
| 1025–1057 | 38 | 87 | 74.5 [b] |

N = number of seedlings. HGP means in the same column and variable (i.e., origin, aspect and altitude) followed by different letters are statistically different ($p \leq 0.05$, Duncan's test).

In north-facing transects, seedlings presented significantly greater height growth (94.0%) compared to south-facing (74.5%) ones. Concerning the transect altitude effect (Table 4), it seems that both the highest and lowest altitudes significantly reduced seedling HGP compared to transects in medium altitudes.

No interaction between the pairs of aspect–provenance, elevation–provenance, and aspect–elevation was detected by applying linear models. The soil parent material and slope variables did not add to any model and were therefore not included in the final model, either.

### 3.3.2. Seedling Crown Diameter

The analysis of variance revealed that seedling provenance ($p \leq 0.001$), transect aspect ($p \leq 0.05$), and altitude ($p \leq 0.001$) significantly affected seedling crown diameter growth during the 3-year monitoring period (Table 5), although not all origins are represented in all experiments, as mentioned above. The overall crown diameter of Parnitha seedlings was more than twice (207.9%) greater than that of Vytina provenance (96.8%) (Table 6). The mean percentage change in crown diameter (CGP) during the 3-year monitoring period in relation to seedling provenance is presented in Figure 5. During this period, the crown diameter growth of Parnitha seedlings was greater than that of Vytina ones. However, during the last summer of the monitoring period, a high percentage (29.8%) of seedlings were found damaged (data not shown). Although this percentage was the lowest calculated during the monitoring periods, the only negative growth value observed in the Parnitha seedlings is probably due, among other factors, to injuries caused by deer which consume the canopy.

**Table 5.** ANOVA table for crown diameter growth of planted *Abies cephalonica* seedlings according to GLM: $Gp_{ijmk} = \mu + o_i + a_j + h_m + o_i \times a_j + e_{ijmk}$.

| Source of Variation | Degrees of Freedom | Mean Square | F | Sig. |
|---|---|---|---|---|
| Provenance | 1 | 828,132.96 | 59.795 | *** |
| Aspect | 1 | 49,375.34 | 3.565 | * |
| Altitude | 2 | 147,401.08 | 10.643 | *** |

*** $p \leq 0.001$; * $p \leq 0.05$.

**Table 6.** The overall mean percentage change in crown diameter (CGP) of planted *Abies cephalonica* seedlings at the end of the monitoring period (autumn 2015), in relation to seedling provenance, transect aspect, and altitude. Different letters represent statistically significant differences within the same column ($p \leq 0.05$, Duncan's test).

| Provenance | N | | CGP |
|---|---|---|---|
| Parnitha | 164 | | 207.9 [a] |
| Vytina | 131 | | 96.8 [b] |
| | **Parnitha** | **Vytina** | |
| | **Aspect** | | |
| | N | N | |
| North | 78 | 44 | 156.4 [b] |
| East | 48 | - | 215.1 [a] |
| South | 38 | 87 | 139.1 [b] |
| | **Altitude** | | |
| | N | N | |
| 926–958 | 39 | - | 173.7 [a] |
| 959–991 | 57 | - | 171.9 [a] |
| 992–1024 | 30 | 44 | 173.3 [a] |
| 1025–1057 | 38 | 87 | 139.1 [b] |

N = number of seedlings. CGP means in the same column and variable (i.e., origin, aspect and altitude) followed by different letter are statistically different ($p \leq 0.05$, Duncan's test).

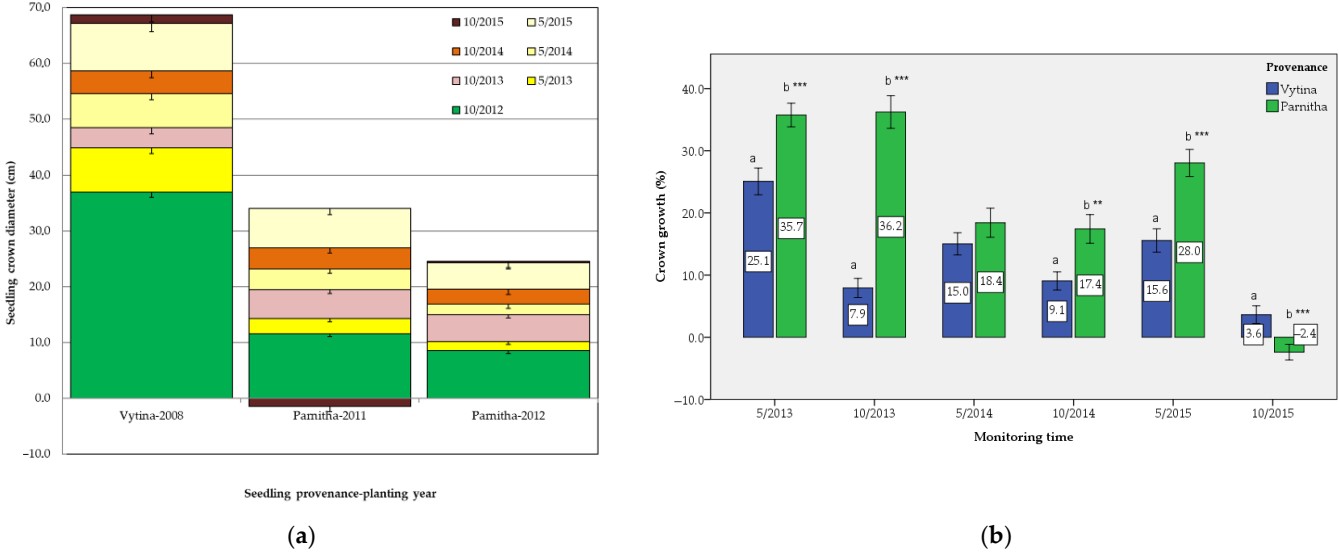

(**a**)          (**b**)

**Figure 5.** Crown diameter (cm) of planted *Abies cephalonica* seedlings (**a**) and the overall mean percentage change in crown diameter (%) (**b**) during the 3-year monitoring period in relation to seedling provenance. Crown diameter growth means in the same monitoring time followed by different letters are statistically different at $p \leq 0.01$ ** and $p \leq 0.001$ *** according to Duncan's test.

Seedlings planted in east-facing transects showed an average crown diameter growth of 215.1% and significantly exceeded those in the north (156.4%) and south (139.1%) ones, while the last two aspects did not show significant differences concerning the crown diameter growth (Table 6). Concerning the altitude effect, it seems that the higher the transect altitude, the less the crown diameter growth. The highest crown growth (173.7%) was observed at the lowest altitudinal zone (926–958 m), although no statistically significant differences were observed among the three-altitudinal zones, i.e., 926–958 m, 959–991 m, and 992–1024 m. In contrast, transects in the highest altitudinal zone (1025–1057 m) showed the lowest crown diameter growth (139.1%).

No interaction between the pairs of aspect–provenance, elevation–provenance, and aspect–elevation was detected by applying linear models. The soil parent material and slope variables did not add to any model and were not included in the final model, either.

### 3.3.3. Crown Condition and Apical Bud

In the last monitoring period (autumn of 2015), the percentage of seedlings with an intact crown, i.e., unharmed, not grazed or cut out branches, were 70.2%, and the percentage of seedlings presenting an apical bud were 87.8% (Table 2). However, seedling provenance greatly affected the percentage of intact crowns. Seedlings from Vytina had 49.6% intact crowns, while those from Parnitha had a significantly greater percentage (86.6%) (Table 7). By contrast, the apical bud presence showed no statistically significant differences between the two provenances and ranged from 83.2% to 91.5% for Vytina and Parnitha seedlings, respectively.

**Table 7.** Crown condition and apical bud presence in planted *Abies cephalonica* seedlings at the end of the monitoring period (autumn 2015), in relation to seedling provenance.

| Variable | Origin | N | Mean | * S.E. |
|---|---|---|---|---|
| Intact seedling crown (%) | Vytina | 131 | 49.6 [a] | 4.39 |
| | Parnitha | 164 | 86.6 [b] | 2.67 |
| Apical bud (%) | Vytina | 131 | 83.2 [a] | 3.28 |
| | Parnitha | 164 | 91.5 [a] | 2.19 |

\* SE: standard error. Means within the same variable (i.e., intact seedling crown and apical bud) followed by different letter are statistically different at $p \leq 0.05$ according to Duncan's test.

Crown condition and apical bud presence were highly affected by both transect altitude and aspect (Figure 6a,b). The highest percentage of seedlings with damaged crowns (70.7%), namely, seedlings observed with destroyed or cut branches, was recorded on the north transects, in the altitudinal zone of 992–1024 m, followed by the east transects in the same altitude (42.4%) (Figure 6a). Seedlings with the most damaged apical buds (44.3%) were also found on the north transects in the altitudinal zone of 992–1024 m (Figure 6b).

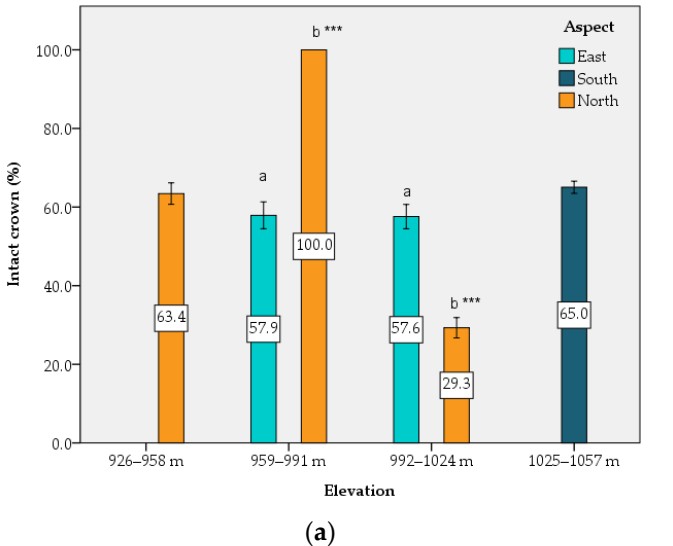

**(a)**

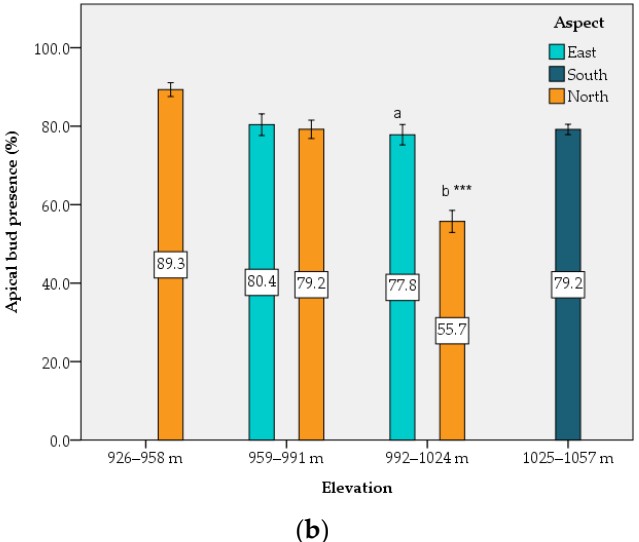

**(b)**

**Figure 6.** Mean intact crown percentage (%) of *Abies cephalonica* planted seedlings (**a**) and apical bud presence (%) (**b**) in autumn 2015, in relation to transect aspect and altitude. Intact crown percentage and apical bud presence means in the same elevation level followed by different letters are statistically different at $p \leq 0.001$ *** according to Duncan's test.

## 4. Discussion

Our results suggest that the performance of Greek fir plantings in a post-fire environment, in the area of Parnitha National Park, were quite successful and presented an overall relatively high survival rate seven years after the first planting period (2008) and eight years after the fire event. Similarly, the seedlings presented satisfactory height and crown diameter growth compared to the baseline values (autumn 2012). A rather high percentage of seedlings having damaged branches and crowns was found, while the damage on seedling apical buds was much lower at the end of the study period. The above results interestingly indicate that although the seedlings were planted in a post-fire forest environment, without being protected by a forest canopy cover, their performance was quite satisfactory. This seems to be contradictory to the species' ecophysiological attitudes, since it is characterized as shade-tolerant and semi-drought-tolerant [39], and its regeneration in forest management systems is performed within an interior forest environment, under the mother stand canopy cover [15–17]. The rather high field performance of the Greek fir seedlings could be attributed to the fact that these were watered during the post-planting period, especially in the first 3–4 years following reforestations and during the summer dry period. Therefore, a correlation analysis between growth or survival with climatic data was not attempted. Watering seems to play a crucial role in seedling survival and growth, by covering them with the necessary moisture for well-balanced water status and temperature fluctuation during the hot and dry summer months. Taking into consideration the fact that *A. cephalonica* can grow in a sub-humid climate characterized by a relatively low annual precipitation of between 700 and 800 mm [40], a watering regime during the crucial post-planting summer months should be considered necessary in forest practice for effective seedling survival and growth.

Overall, our results indicate that *A. cephalonica* seedlings from the two different provenances (origins) respond quite similarly after reforestations. Higher survival rates were achieved by the seedlings originating from Vytina, while seedlings from Parnitha exhibited higher growth. This could be attributed to the fact that the summer drought periods affected the mortality of Parnitha seedlings more, while Vytina seedlings were slightly affected, since the latest provenance was mainly planted in favored sites and aspects. In addition, we can assume that the lower survival rate of Parnitha seedlings is interpreted in the reforestation period and year, 1–2 years before the onset of monitoring in the present study. According to Ganatsas et al. [6], Vytina seedlings seemed to overcome stress 2 years after planting and adapted to the local post-fire conditions and were well established (with a mean survival rate of 65.3%). However, these results preceded the examined monitoring period, thus explaining the high and almost stable survival rate observed in this study. A similar survival rate (61.5%) was recorded by other researchers for the same provenance 2 years after the reforestation [26]. Lower survival percentages were recorded 3 and 4 years after planting, 52.7% and 46.2%, respectively [26].

For the Mediterranean forest plantations, surviving summer water stress after planting during the early years presents a major challenge [41]. Detsis et al. [25] also reported that water availability is among the basic factors affecting Greek fir seedlings' performance in the area. This response to the drought period could be linked either to genetic adaptations of individual Greek fir provenances or the planting stock quality; planting stock from Vytina may have been more hardened before and at the time of planting. In addition, wildfire characteristics exert a strong influence on stand characteristics [42], and the variability throughout the landscape of both fire intensity and severity produces heterogeneous post-fire environments [43], thus affecting plant establishment [42,44,45]. However, any possible effect of wildfire severity and intensity on seedlings' performance was not studied, since there were no available spatial data for the wildfire, and the wildfire completely destroyed the forest in the burnt area, meaning that no spatial differentiations were observed.

Furthermore, seedling survival was slightly higher in north-facing slopes followed by the south-facing ones, while it was lower in east-facing transects. This may be explained as planting was performed immediately after fire, and under the great pressure imposed by



citizens for emergency reforestation. Planting started mainly in north-facing slopes, and thus, seedlings were well adapted to local microenvironments. Contrary to the common belief that northern aspects are much more favorable than southern ones, the transect aspect did not statistically influence seedling survival rate significantly, as had been reported by other studies [46]. Moreover, elevation and aspect affect the variability of fire severity [45], which in turn affects the survival of seedlings. In general, in divergent Mediterranean environments, aspect affects survival in varying ways, resulting in mixed conclusions [47].

During the monitoring period, the height and crown diameter growth of Parnitha seedlings was significantly greater than that of Vytina's and thus adapted better to the local site conditions. Species adaptation to local conditions is a well-known mechanism in plant ecology [48,49] which is highly taken into consideration during planting for ecosystem restoration [50].

Site heterogeneity remarkably affected seedling performance [51]. Among the studied environmental factors, the aspect and the altitude seemed to play an important role in seedling growth. At the end of the monitoring period, a greater growth of Greek fir seedlings, both in height and crown diameter, was observed in the north- and east-facing transects, respectively, and in medium altitudes (959–1024 m), probably due to their ability to create more favorable temperatures for plant growth [52], especially under dry climatic conditions [53]. This growth pattern is common in forest ecosystems in the northern hemisphere [52,54]; however, the interaction with other environmental variables should be considered. In general, seedling performance is affected by the microenvironments [55]. By contrast, the slope and type of bedrock (soil parent material) did not affect seedling growth.

The percentage of intact crown and the apical bud presence in the seedlings were both highly affected by altitude and aspect. The greater percentage of seedlings with damaged crowns and the lower apical bud presence were recorded in north-facing transects and the medium altitudinal zone (992–1024 m). This can be explained as the local population of red deer thrives mainly in the north-facing slopes, since these sites are the most isolated from human presence. This isolation favors deer movement and activity.

Taking into consideration the negative impact of the current climate change, as predicted by several scenarios, it is expected that many forest species could be at high risk from its negative impact [56]. Under these changing environmental conditions, endemic species, such as the studied Greek fir, may not be able to adjust to these changes and adapt and may therefore be the first to go extinct [3]. For example, the inability of the species to regenerate naturally after fire may lead to a secondary ecological succession to other (degraded) plant communities [6]. Native forests may change in composition, and some species may be entirely eliminated over large areas as a result of climatic change. Conservation of genetic resources will be necessary to restore declining forests; even in a seed source that is not adapted to the plantation site, some seedlings will survive and grow remarkably well [57].

The early establishment and abundance of shade-tolerant conifer species, as was (partly) done in Parnitha National Park, contrasts with traditional stand development models which need significant time for monitoring, evaluation, resource availability administration, and knowledge and might have long-term impacts on biodiversity, landscapes, and livelihoods [24]. Recruitment following large-scale disturbance is assumed to take decades, if not centuries [58]. Due to the lack of an official and universal approach on forest restoration, imposing principles and standards can increase the effectiveness of restoration processes across different ecosystems, locally or globally [59]. Long-term reforestation success in areas which are highly vulnerable to climate change depends on using plant material with appropriate levels of genetic diversity [60], with local or regional genetic variation, which ensures the survival and resilience of a planted forest [24]. Collecting seeds from native trees belonging to different local provenances across the parent population is an excellent practice [61–63] that ensures adequate genetic diversity determined by the size of the parental population [63] and collection rules [64]. However, it may be prudent to include some germplasm of the same species from a predicted "future climate", namely, a region with a climate similar to that predicted for the area being restored [59]. In addi-

tion, population size is probably one of the most important criteria, useful for indicating both genetic erosion and genetic pollution [65] as well as being an essential element for future evolution [66]. Trees opt to make rapid adaptive changes while maintaining a high level of genetic diversity within the population. Their genetic diversity along with the environmental conditions influence the result of combined evolutionary forces [67].

Thus, the conservation of biodiversity within the National Park of Parnitha enforces the need for planting *A. cephalonica* seedlings. The findings of the study aspire to mitigate the risk of the important endemic Mediterranean fir *A. cephalonica* extinction. However, further research data are urgently needed to secure species conservation and sustainability from a long-term perspective.

## 5. Conclusions

Seed and, consequently, genetic material conservation are important either for the natural regeneration process or for artificial plantations, even for the restoration of native forests. Appropriate plant species, along with their suitable genetic provenances, are critical to establishing reforested areas with high vulnerability to climate change. To the best of our knowledge, this is the first study examining the longevity and performance of *Abies cephalonica* seedlings, originating from two provenances, selected for extensive reforestations after a wildfire. *A. cephalonica* reforestations constitute a "living experiment" in the study area (Parnitha National Park, Attica, Greece), which is included in the European Natura 2000 network, hosting both the endemic tree and its forest. Both provenances showed a relatively successful survival rate, with seedling mortality occurring during the first 1–2 years following planting, mainly during the summer dry period. Furthermore, planted seedlings, from both origins, presented a satisfactory growth in height and crown diameter with the seedlings of local origin (Mt. Parnitha) preceding. Seedling growth continues not only during the growth period (summer) but also almost during the whole year. Given the fact that restoration of degraded forest ecosystems is a complicated task, especially in post-fire conditions, further research on planted seedling survival and growth monitoring is proposed, enhanced by long-term data.

**Author Contributions:** K.I. and E.N.D. conceived the initial outline of the article; K.I. led the writing and prepared the figures; M.T. and P.G. contributed to writing—original draft preparation; K.I., K.K., and E.N.D. contributed to fieldwork data collection; P.G. contributed to methodology, data elaboration, review, and editing; E.N.D. coordinated the project implementation and author contributions. All authors contributed to writing, developing, and reviewing the manuscript. All authors have read and agreed to the published version of the manuscript.

**Funding:** This research received funding in the framework of the national research project "Contribution to the post-fire management of Parnitha National Park", assigned by the Hellenic Ministry of Environment and Energy, General Directorate of Development and Forest Protection and Agro environment and funded by the Green Fund (2012–2015).

**Data Availability Statement:** Not applicable.

**Acknowledgments:** The authors acknowledge the local Forest Department of Parnitha (Region of Attica) and the Management Body of Parnitha NP for the reforested site indication and Despina Paitaridou, Ministry of Environment and Energy—Directorate of Reforestation and Watershed Management, for seedling provenance clarifications. The authors are grateful to Vassiliki Gouma, Asimina Skouteri, Victoria Menteli, and Panagiota-Effrosyni Radaiou for their contribution to field work and Danae Panayiotopoulou, IMFE Librarian MSc, for her additional proofreading services.

**Conflicts of Interest:** The authors declare no conflict of interest.

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
