# Peer review of "Effect of Seedling Provenance and Site Heterogeneity on Abies cephalonica Performance in a Post-Fire Environment"

_sustainability, doi:10.3390/su13116097_

Round 1

Reviewer 1 Report

The article is interesting, but requires more information and modification.

Author Response

We thank Reviewer 1 for the positive and valuable comments and for the targeted observations that greatly improved the manuscript. All the comments were considered in manuscript (see the attached file).

Reviewer 2 Report

This study looks at two different provenances of Abies species growing in a post-fire environment and compares seedling survival, growth, and crown development. Overall, the study is exploratory research to assess survival metrics. The paper, however, has several areas that need improvement. Especially, the methodology section could be improved more. Some of my concerns are:

  1. Authors have used growth rate as a metric for comparison between two provenances (Line 132). Are the growth rates the same over these periods for two provenances (one planted in 2008 and the other in 2011/2012)? Please provide some evidence from the literature to back up your assumption.
  2. Some of the citations are loosely used. For example, in Line 134, what is the point of the citation of research by Peltola et al. 2002. How are 20-year-old Scots pines relevant to your study?
  3. In particular, the altitudinal variation in this study is rather small (about 130 m). Was there a biological basis that would warrant the reclassification in four zones? I also find this a little confusing when looking at some of the results in tables 4 and 6 where the number of seedlings in zone 1025-1057 is almost twice that of the other zones. More importantly, Vytina has been planted only above 1000m in this study? These tables need to be modified to include the number of seedlings by provenances for each altitudinal group and their mean values for CGP or HGP.
  4. The statistical analysis section should be improved to include model information (e.g., what the predictor variables are; see Line 163)
  5. Some paragraphs in the discussion section need revision. For instance, Line 410-434 is rather generic. It should be rewritten to make it concise and highlight the applicability of the results of this provenance comparison.

    Line comments:

    Table 2: Include seedling numbers as well. This will help answer many of the questions that may arise later in the manuscript.

    Line 185-186: Is this result showing one of the provenances that were planted in 2008.

    Line 187-188: Not clear what the sentence about initial relevant values means?

    Line 207-213: This paragraph rather seems abrupt. Present your model in the methods and be specific about what your predictor variables are?

    Section 3.3: Move this entire section to methods.

    Table 4 and Table 6: Please see my earlier comments about the number of seedlings for provenances within each altitudinal zone.

    Line 266: Not all provenances are represented in (e.g. altitudinal classes- mainly Vytina is mostly planted at >1000 m see Table 1). Altitudinal differences observed might likely be an artifact of the study design. Please clarify.

    Line 272-273: Is this damage mentioned different from what is presented in the crown condition section? Would the tree height not affect the level of deer damage in these stands?

    Line 301: Please mention which values are for which provenances.

    Line 453: What do you mean by satisfactory growth? What was the metric/reference to assess it?

Author Response

We thank reviewer 2 for the positive comment and careful observations that considerably improved the manuscript. All the comments were considered in manuscript (see the attached file).

Round 2

Reviewer 1 Report

Thank you for your efforts.

Author Response

We thank reviewer 1 for the positive comments  that improved the relevant sections of the manuscript. 

Reviewer 2 Report

I thank the authors for addressing my prior concerns. The manuscript now merits publication in the journal. See below for minor comments:

a) Tables 4 and 6 seem a bit off and difficult to follow. Please rearrange these tables.

b) Figure 2 caption needs editorial correction for 'twice annually,..... every year.'

Author Response

We thank reviewer 2 for the positive comments and the 2 careful observations that improved the relevant sections of the manuscript. All new comments were considered in revised manuscript.